# Post-Hoc Robustness Enhancement in Graph Neural Networks with Conditional Random Fields

## Abstract

Graph Neural Networks (GNNs), which are nowadays the benchmark approach in graph representation learning, have been shown to be vulnerable to adversarial attacks, raising concerns about their real-world applicability. While existing defense techniques primarily concentrate on the training phase of GNNs, involving adjustments to message passing architectures or pre-processing methods, there is a noticeable gap in methods focusing on increasing robustness during inference. In this context, this study introduces RobustCRF, a post-hoc approach aiming to enhance the robustness of GNNs at the inference stage. Our proposed method, founded on statistical relational learning using a Conditional Random Field, is model-agnostic and does not require prior knowledge about the underlying model architecture. We validate the efficacy of this approach across various models, leveraging benchmark node classification datasets.

## 1 Introduction

Deep Neural Networks (DNNs) have demonstrated exceptional performance across various domains, including image recognition, object detection, and speech recognition (Chen et al., 2019; Ni et al., 2022; Shim et al., 2021). The growing interest in handling irregular and unstructured data, particularly in domains like bioinformatics, has pushed some attention toward graph-based representations. Graphs have emerged as the preferred format for representing such irregular data due to their ability to capture interactions between elements, whether individuals in a social network or interactions between atoms. In response to this need, Graph Neural Networks (GNNs) (Kipf & Welling, 2017; Veličković et al., 2017; Xu et al., 2019b) have been proposed as an extension of DNNs tailored to graph-based data. GNNs excel at generating meaningful representations for individual nodes by leveraging both the graph's structural information and its associated features. This approach has demonstrated significant success in addressing challenges such as protein function prediction (Gilmer et al., 2017), materials modeling (Duval et al., 2023), and recommendation systems (Wu et al., 2019b).

Alongside their achievements, these deep learning-based approaches have exhibited vulnerability to various data alterations, including noisy, incomplete, or out-of-distribution examples (Günnemann, 2022). While such alterations may naturally occur in data collection, adversaries can deliberately craft and introduce them, resulting in adversarial attacks. These attacks manifest as imperceptible modifications to the input that can deceive and disrupt the classifier. In these attacks, the adversary's objective is to introduce subtle noise in the features or manipulate some edges in the graph structure to alter the initial prediction made by the input graph. Depending on the attacker's goals and level of knowledge, different attack settings can be considered. For instance, poisoning attacks (Zügner & Günnemann, 2019) involve manipulating the training data to introduce malicious points, while evasion attacks (Dai et al., 2018) focus on attacking the model during the inference phase without further model adaptation.

Given the susceptibility of GNNs to adversarial attacks, their practical utility is constrained. Therefore, it becomes imperative to study and enhance their robustness. Various defense strategies have been proposed to mitigate this vulnerability, including pre-processing the input graph (Entezari et al., 2020; Wu et al., 2019a), edge pruning (Zhang & Zitnik, 2020), and adapting the message passing scheme

(Zhu et al., 2019; Liu et al., 2021; Abbahaddou et al., 2024). Most of these defense methods operate by modifying the underlying model architecture or the training procedure, i.e., focusing on the training side of GNNs and, therefore, limiting their applicability. This limitation is especially pertinent when dealing with pre-trained models, which have become the standard for real-world applications. Additionally, modifying the model architecture during training poses the risk of degrading accuracy on clean, non-attacked graphs, and usually, these modifications are not adapted for all possible architectures.

In light of these challenges, our work introduces a post-hoc defense mechanism, denoted as RobustCRF, aimed at bolstering the robustness of GNNs during the inference phase using statistical relational learning. RobustCRF is model-agnostic, requiring no prior knowledge of the underlying model, and is adaptable to various architectural designs, providing flexibility and applicability across diverse domains. Central to our approach is the assumption that neighboring points in the input manifold, accounting for graph isomorphism, should yield similar predictions in the output manifold. Based on our robustness assumption, we employ a Conditional Random Field (CRF) (Lafferty et al., 2001) to adapt and edit the model's output to preserve the similarity relationship between the input and output manifold.

While the proposed approach is applicable to a wide range of tasks and models, including those in the computer vision domain, our primary focus revolves around Graph Neural Networks (GNNs). We start by introducing our CRF-based post-hoc robustness enhancement model. Recognizing the potential complexity associated with this task and aiming to deliver a robustness technique that remains computationally affordable, we study a sampling strategy for both the discrete space of the graph structure and the continuous space of node features. We finally proceed with an empirical analysis to assess the effectiveness of our proposed technique across various models, conducting also a comprehensive examination of the parameters involved. In summary, our contributions can be outlined as follows: **(1)** model-agnostic robustness enhancement: we present RobustCRF, a post-hoc approach designed to enhance the robustness of underlying GNNs, without any assumptions about the model's architecture. This feature underscores its versatility, enabling its application across diverse models; **(2)** theoretical underpinnings and complexity reduction: we conduct a comprehensive theoretical analysis of our proposed approach and enhance our general architecture through the incorporation of sampling techniques, thereby mitigating the complexity associated with the underlying model.

## 2 RELATED WORK

**Attacking GNNs.** A multitude of both poisoning and evasion adversarial attacks targeting GNNs models has surged lately (Günnemann, 2022; Zügner et al., 2018). Namely, gradient-based techniques (Xu et al., 2019a), such as Proximal Gradient Descent (PGD), have been employed to tackle the adversarial aim, which is framed as an optimization task aiming to find the nearest adversary to the input point while fulfilling the adversarial objective. Building upon this foundation, Mettack (Zügner & Günnemann, 2019) extends the approach by formulating the problem as a bi-level optimization task and harnessing meta-gradients for its solution. In a different perspective, Nettack (Zügner et al., 2018) introduces a targeted poisoning attack strategy, encompassing both structural and node feature perturbations, employing a greedy optimization algorithm to minimize an attack loss with respect to a surrogate model. Diverging from these classical search problems, Dai et al. (2018) approach the adversarial search task through the lens of Reinforcement Learning techniques.

**Defending GNNs.** Different defense strategies have been proposed to counter the previously presented attacks on GNNs. GNN-SVD (Entezari et al., 2020) employs low-rank approximation of the adjacency matrix to filter out noise, while similar pre-processing techniques are used by GNN-Jaccard (Wu et al., 2019a) to identify potential edge manipulations. Additionally, methods like edge pruning (Zhang & Zitnik, 2020) and transfer learning (Tang et al., 2020) have been used to mitigate the impact of poisoning attacks. Notably, most research efforts have predominantly focused on addressing structural perturbations, with relatively fewer strategies developed to counter attacks targeting node features. For instance, in (Seddik et al., 2022), the inclusion of a node feature kernel within message passing was proposed to reinforce GCNs. RobustGCN (Zhu et al., 2019) uses Gaussian distributions as hidden node representations in each convolutional layer, effectively mitigating the influence of both structural and feature-based adversarial attacks. Finally, in GCORN

(Abbahaddou et al., 2024), an adaptation of the message passing scheme has been proposed by using orthonormal weights to counter the effect of node feature-based adversarial attacks.

The majority of the previously discussed methods intervene during the model's training phase, necessitating modifications to the underlying architecture. However, this strategy exhibits certain limitations; it may not be universally applicable across diverse architectural designs, and it can potentially result in a loss of accuracy when applied to the clean graph. Furthermore, these methods may not be suitable for scenarios where users prefer to employ pre-trained models, as they necessitate model retraining. Consequently, the need for proposing and crafting post-hoc robustness enhancements becomes increasingly apparent. Unfortunately, the existing methods in this domain remain quite limited. Addressing this gap in the literature, our work aims to contribute to this essential area. One commonly employed post-hoc approach is randomized smoothing (Bojchevski et al., 2020; Carmon et al., 2019), which involves injecting noise into the inputs at various stages and subsequently utilizing majority voting to determine the final prediction. Note that randomized smoothing has actually been initially borrowed from the optimization community (Duchi et al., 2012). Despite its popularity, randomized smoothing exhibits several limitations, including suffering from the *shrinking phenomenon* where decision regions shrink or drift as the variance of the smoothing distribution increases (Mohapatra et al., 2021). Other works also identified that the smoothed classifier is *more-constant* than the original model, i.e., it forces the classification to remain invariant over a large input space, resulting a huge drop of the accuracy (Anderson & Sojoudi, 2022; Krishnan et al., 2020; Wang et al.).

## 3 Preliminaries

Before continuing with our contribution, we begin by introducing notation and some fundamental concepts.

### 3.1 Graph Neural Networks

Let $G = (V, E)$ be a graph where $V$ is the set of vertices and $E$ is the set of edges. We will denote by $n = |V|$ and $m = |E|$ the number of vertices and number of edges, respectively. Let $\mathcal{N}(v)$ denote the set of neighbors of a node $v \in V$, i.e., $\mathcal{N}(v) = \{u \colon (v, u) \in E\}$. The degree of a node is equal to its number of neighbors, i.e., equal to $|\mathcal{N}(v)|$ for a node $v \in V$. A graph is commonly represented by its adjacency matrix $\mathbf{A} \in \mathbb{R}^{n \times n}$ where the $(i, j)$-th element of this matrix is equal to the weight of the edge between the $i$-th and $j$-th node of the graph and a weight of $0$ in case the edge does not exist. In some settings, the nodes of a graph might be annotated with feature vectors. We use $\mathbf{X} \in \mathbb{R}^{n \times K}$ to denote the node features where $K$ is the feature dimensionality.

A GNN model consists of a series of neighborhood aggregation layers which use the graph structure and the nodes' feature vectors from the previous layer to generate new representations for the nodes. Specifically, GNNs update nodes' feature vectors by aggregating local neighborhood information. Suppose we have a GNN model that contains $T$ neighborhood aggregation layers. Let also $\mathbf{h}_v^{(0)}$ denote the initial feature vector of node $v$, i.e., the row of matrix $\mathbf{X}$ that corresponds to node $v$. At each iteration ($t > 0$), the hidden state $\mathbf{h}_v^{(t)}$ of a node $v$ is updated as follows:

$$\mathbf{a}_v^{(t)} = \text{AGGREGATE}^{(t)}\Big(\big\{\mathbf{h}_u^{(t-1)} \colon u \in \mathcal{N}(v)\big\}\Big),$$

$$\mathbf{h}_v^{(t)} = \text{COMBINE}^{(t)}\Big(\mathbf{h}_v^{(t-1)}, \mathbf{a}_v^{(t)}\Big),$$

where $\text{AGGREGATE}(\cdot)$ is a permutation invariant function that maps the feature vectors of the neighbors of a node $v$ to an aggregated vector. This aggregated vector is passed along with the previous representation of $v$, i.e., $\mathbf{h}_v^{(t-1)}$, to the $\text{COMBINE}(\cdot)$ function which combines those two vectors and produces the new representation of $v$. After $T$ iterations of neighborhood aggregation, to produce a graph-level representation, GNNs apply a permutation invariant readout function, e.g., the sum operator, to nodes feature as follows:

$$\mathbf{h}_G = \text{READOUT}\Big(\big\{\mathbf{h}_v^{(T)} \colon v \in V\big\}\Big). \tag{1}$$

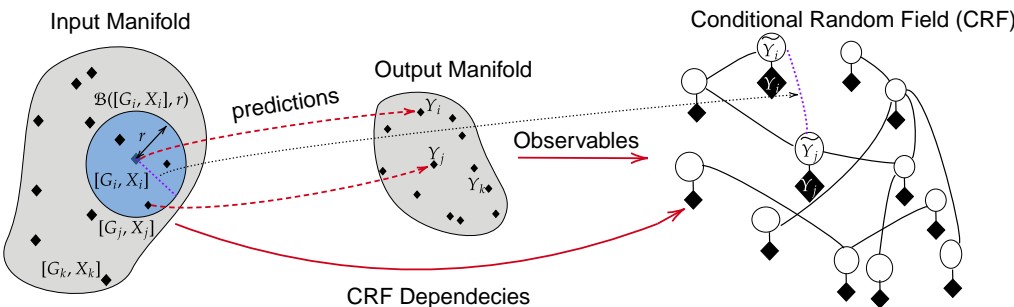

Figure 1: Illustration of RobustCRF. We use input graphs manifold to generate the structure of the CRF, i.e., $V^{CRF}$, $E^{CRF}$. We use the GNN's predictions to generate the observables $\{Y_a \mid a \in V^{CRF}\}$, we then run the CRF inference to generate the new GNN's predictions $\{\tilde{Y}_a \mid a \in V^{CRF}\}$.

## 3.2 CONDITIONED RANDOM FIELDS

A probabilistic graphical model (PGM) is a graph framework that compactly models the joint probability distributions $P$ and dependence relations over a set of random variables $\tilde{Y} = \{\tilde{Y}_1, \ldots, \tilde{Y}_m\}$ represented in a graph. The two most common classes of PGMs are Bayesian networks (BNs) and Markov Random Fields (MRFs) (Heckerman, 2008; Clifford, 1990). The core of the BN representation is a directed acyclic graph (DAG). As the name suggests, DAG can be represented by a graph with no loopy structure where its vertices serve as random variables and directed edges serve as dependency relationships between them. The direction of the edges determines the influence of one random variable on another. Similarly, MRFs are also used to describe dependencies between random variables in a graph. However, MRFs use undirected instead of directed edges and permit cycles. An important assumption of MRFs is the *Markov property* i.e., for each pair of nodes $(a, b)$ such as $e_{ab} \notin E^{MRF}$, the node $a$ is independent of the node $b$ conditioned on $v$'s neighbors:

$$\forall a \in V^{MRF}, \quad \tilde{Y}_a \perp \tilde{Y}_{V \setminus \mathcal{N}(a)} | \mathcal{N}(a).$$

An important special case of MRFs arises when they are applied to model a conditional probability distribution $P(\tilde{Y}|Y, V^{MRF}, E^{MRF})$, where $Y = \{Y_1, \ldots, Y_m\}$ are additional observed node features. These types of graphs are called Conditioned Random Fields (CRFs) (Sutton et al., 2012; Wallach et al., 2004). Formally, a CRF is Markov network over random variables $\mathcal{Y}$ and observation $\mathcal{X}$, the conditional distribution is defined as follows:

$$P(\tilde{Y}|Y, V^{CRF}, E^{CRF}) = \frac{1}{Z(Y, V^{CRF}, E^{CRF})} \exp\left\{-\sum_{a \in V^{CRF}} \phi_a(\tilde{Y}_a) - \sum_{(a,b) \in E^{CRF}} \phi_{ab}(\tilde{Y}_a, \tilde{Y}_b)\right\},$$

where $Z(\mathcal{X}, E)$ is the partition function, and $\phi_v(Y_v, X, E)$ and $\phi_{vu}(Y_v, Y_u, X, E)$ are potential functions contributed by each node $v$ and each edge $(u, v)$ and usually defined as simple linear functions or learned by simple regression on the features (e.g., logistic regression).

## 4 ROBUSTCRF: A CRF-BASED ROBUSTNESS ENHANCEMENT MODEL

### 4.1 ADVERSARIAL ATTACKS

Let us consider our graph space $(\mathcal{G}, \|\cdot\|_{\mathcal{G}})$, feature space $(\mathcal{X}, \|\cdot\|_{\mathcal{X}})$, and the label space $(\mathcal{Y}, \|\cdot\|_{\mathcal{Y}})$ to be measurable spaces. Given a GNN $f : (\mathcal{G}, \mathcal{X}) \to \mathcal{Y}$, as introduced in Section 3.1, an input data point $G \in \mathcal{G}$ and its corresponding prediction $y \in \mathcal{Y}$ where $f(G) = y$, the goal of an adversarial attack is to produce a perturbed graph $\tilde{G}$ slightly different from the original graph with its predicted class being different from the predicted class of $G$. This could be formulated as finding a $\tilde{G}$ with $f(\tilde{G}) = \tilde{y} \neq y$ subject to $d(G, \tilde{G}) < \epsilon$, with $d$ being some distance function between the original

and perturbed graphs. This could be a distance taking into account both the graph structure, in terms of the adjacency matrix, and the corresponding node features, defined as,

$$d_{\alpha,\beta}([A, X], [\tilde{A}, \tilde{X}]) = \min_{P \in \Pi}\{\alpha\|A - P\tilde{A}P^\top\|_2 + \beta\|X - P\tilde{X}\|_2\},$$

The distance $d_{\alpha,\beta}$ has already been used in the literature (Abbahaddou et al., 2024). This distance is advantageous because it incorporates both the graph structure and the node features. The matrix $P$ corresponds to a permutation matrix used to order nodes from different graphs. By using Optimal Transport, we find the minimum distance over the set of permutation matrices, which corresponds to the optimal matching between nodes in the two graphs.

## 4.2 MOTIVATION

There are different theoretical definitions of robustness (Cheng et al., 2021; Weng et al., 2018), but they all rely on one assumption: *if two inputs are adjacent in the input space, their predictions should be adjacent in the output space*. For example, most existing approaches measure the robustness of a neural network via the *Attack Success Rate* (ASR), the percentage of attack attempts that produce successful adversarial examples, which implies that close input data should be predicted similarly (Wu et al., 2021; Goodfellow et al., 2014). Additionally, there are also works that use the neural networks distortion as a robustness metric (Cheng et al., 2021; Weng et al., 2018; Carlini & Wagner, 2016). Intuitively, large distortion implies potentially poor adversarial robustness since a small perturbation applied to these inputs will lead to significant changes in the output. Consequently, most robustness metrics measure the extent to which the network's output is changed when perturbing the input data, indicating the network's vulnerability to adversarial attacks, which is directly equivalent to our assumption. To respect this assumption, we construct a CRF, where the node set $V^{CRF}$ represents the set of possible GNN inputs $\mathcal{G} \times \mathcal{X}$, while the CRF edge set $E^{CRF}$ represents the set of pair inputs $([G, X], [\tilde{G}, \tilde{X}])$ such that $[\tilde{G}, \tilde{X}]$ belongs to the ball $\mathcal{B}$ of radius $r > 0$ surrounding $[G, X]$, as follows:

$$\mathcal{B}([G, X], r) = \left\{ \left[\tilde{G}, \tilde{X}\right] : d^{\alpha,\beta}([G, X], [\tilde{G}, \tilde{X}]) \leq r \right\}.$$

Our method examines how the model performs more generally within a defined neighborhood, instead of considering the model's behavior only under individual adversarial attacks. This perspective leans towards a concept of *average* robustness, expanding on the conventional worst-case based adversarial robustness that is often emphasized in adversarial studies (Abbahaddou et al., 2024). A similar *average robustness* concept was studied and showed to be appropriate for computer vision (Rice et al., 2021).

## 4.3 MODELING THE ROBUSTNESS CONSTRAINT IN A CRF

Let $Y_a$ be the output prediction of the trained GNN $f$ on the input $a = [G, X] \in V^{CRF}$. The main goal of using a CRF is to update the predictions $Y = \{Y_a | a \in V^{CRF}\}$ into new predictions $\tilde{Y} = \{\tilde{Y}_a | a \in V^{CRF}\}$ that respect the *robustness assumption*. To do so, we model the relation between the two predictions $Y$ and $\tilde{Y}$ using a CRF, maximizing the following conditional probability:

$$P(\tilde{Y}|Y, V^{CRF}, E^{CRF}) = \frac{1}{Z} \exp \left\{ -\sum_{a \in V^{CRF}} \phi_a(\tilde{Y}_a, Y_a) - \sum_{b \text{ s.t. } (a,b) \in E^{CRF}} \phi_{ab}(\tilde{Y}_a, \tilde{Y}_b) \right\}, \quad (2)$$

where $Z$ is the partition function, $\phi_a(\tilde{Y}_a, Y_a)$ and $\phi_{ab}(\tilde{Y}_a, \tilde{Y}_b)$ are potential functions contributed by each CRF node $a$ and each CRF edge $(a, b)$ and usually defined as simple linear functions or learned by simple regression on the features (e.g., logistic regression). The potential functions $\phi_a$ and $\phi_{ab}$ can be either fixed or trainable to optimize a specific objective. In this paper, we define the potential functions as follows:

$$\begin{cases} \phi_a(\tilde{Y}_a, Y_a) = \sigma\|\tilde{Y}_a - Y_a\|_2^2, \\ \phi_{ab}(\tilde{Y}_a, \tilde{Y}_b) = (1 - \sigma)g_{ab}\|\tilde{Y}_a - \tilde{Y}_b\|_2^2, \end{cases}$$

where $g_{ab}$ denotes the similarity between two inputs $a = [G, X]$ and $b = [\tilde{G}, \tilde{X}]$. Representing GNN inputs in the CRF can be seen as a constraint problem to enhance the robustness: Finding a new prediction $\tilde{Y}_a$ that meets the demands of the robustness objective, and at the same time stays as close

as possible to the original prediction of the GNN, $Y_a = f(a)$ where $a = [G, X]$ is a GNN input. The parameter $\sigma$ is used to adjust the importance of the two potential functions. In Figure 1, we illustrate the main idea of the proposed RobustCRF model.

Now that we have defined the CRF and its potential functions, generating the new prediction $\{\tilde{Y}_p | p \in V^{CRF}\}$ is intractable for two reasons. First, the partition function $Z$ is usually intractable. Second, the CRF distribution $P$ represents a potentially infinite collection of CRF edges $E^{CRF}$. We need, therefore, to derive a tractable algorithm to generate the smoothed prediction $\tilde{Y}$. In what follows, we show how to overcome these two challenges.

### 4.4 MEAN FIELD APPROXIMATION

We aim to derive the most likely $\tilde{Y}$ from the initial distribution $P$ defined in Eq. 2, as:

$$\tilde{Y}^* = \arg\max P(\tilde{Y}|Y, V^{CRF}, E^{CRF}). \tag{3}$$

Since the inference is intractable, we used a Variational Inference method where we propose a family of densities $\mathcal{D}$ and find a member $Q^* \in \mathcal{D}$ which is close to the posterior $P(\tilde{Y}|Y, V^{CRF}, E^{CRF})$. Thus, we approximate the initial task in Equation 3 with a new objective:

$$\begin{cases} \tilde{Y}^* = \arg\max_{\tilde{Y}} Q^*(\tilde{Y}), \\ Q^* = \arg\min_{Q \in \mathcal{D}} \mathcal{KL}(Q|P). \end{cases} \tag{4}$$

The goal is to find the distribution $Q^*$ within the family $\mathcal{D}$, which is the closest to the initial distribution $P$. In this work, we used the *mean-field approximation* (Blei et al., 2016) that enforces full independence among all latent variables. Mean field approximation is a powerful technique for simplifying complex probabilistic models, widely used in various fields of machine learning and statistics (Andrews & Baguley, 2017; Wang & Blei, 2013). Thus, the variational distribution over the latent variables factorizes as:

$$\forall Q \in \mathcal{D}, \ Q(\tilde{Y}) = \prod_{a \in V^{CRF}} Q_a(\tilde{Y}_a). \tag{5}$$

The exact formula of the optimal surrogate distribution $Q$ can be obtained using Lemma 4.1. We will use Coordinate Ascent Inference (CAI), iteratively optimizing each variational distribution and holding the others fixed.

**Lemma 4.1.** *By solving the system of Eq. 4, we can get the optimal distribution $Q^*$ as follows:*

$$\forall a \in V^{CRF}, \ Q_a(\tilde{Y}_a) \propto \exp\left\{\mathbb{E}_{-a}\left[\log P\left(\tilde{Y}|Y, V^{CRF}, E^{CRF}\right)\right]\right\}. \tag{6}$$

The proof of Lemma 4.1 is provided in Appendix B. CAI algorithm iteratively updates each $Q_a(\tilde{Y}_a)$. The ELBO converges to a local minimum. Using Equations 2 and 6, we get the optimal surrogate distribution as follows:

$$\forall a \in V^{CRF}, \ Q(\tilde{Y}_a) \propto \exp\left\{\sigma\|\tilde{Y}_a - Y_a\|_2^2 + (1-\sigma)\sum_{b \text{ s.t. } (a,b) \in E^{CRF}} g_{ab}\|\tilde{Y}_a - \tilde{Y}_b\|_2^2\right\}. \tag{7}$$

Thus, for each GNN input $a \in V^{CRF}$, $Q_a(\tilde{Y}_a)$ is a Gaussian distribution that reaches the highest probability at its expectation:

$$\forall a \in V^{CRF}, \ \arg\max Q_a = \frac{\sigma Y_a + (1-\sigma)\sum_{b \text{ s.t. } (a,b) \in E^{CRF}} g_{ab}\tilde{Y}_b}{\sigma + (1-\sigma)\sum_{b \text{ s.t. } (a,b) \in E^{CRF}} g_{ab}}. \tag{8}$$

Using the CAI algorithm, we can thus utilize the following update rule:

$$\forall a \in V^{CRF}, \ \tilde{Y}_a^{k+1} = \frac{\sigma Y_a + (1-\sigma)\sum_{b \text{ s.t. } (a,b) \in E^{CRF}} g_{ab}\tilde{Y}_b^k}{\sigma + (1-\sigma)\sum_{b \text{ s.t. } (a,b) \in E^{CRF}} g_{ab}}. \tag{9}$$

### 4.5 REDUCING THE SIZE OF THE CRF

The number of possible inputs is usually very large for discrete data and infinite for continuous data. This could make the inference intractable due to the potential large size of $E^{CRF}$. Therefore, instead of considering all the possible CRF neighbors of an input $a$, i.e., all inputs $b \in V^{CRF}$ such that $d^{\alpha,\beta}(a, b) \leq r$, we can consider only a subset of $L$ neighbors by randomly sampling from the CRF neighbors of $p$. The update rule in Eq. 9 becomes:

$$\tilde{Y}_a^{k+1} = \frac{\sigma Y_a + (1-\sigma) \sum_{b \in \mathcal{U}^L(a)} g_{ab} \tilde{Y}_b^k}{\sigma + (1-\sigma) \sum_{b \in \mathcal{U}^L(a)} g_{ab}}, \tag{10}$$

where $\mathcal{U}^L(a)$ denotes a set of $L$ randomly sampled CRF neighbors of a graph $[G, X]$. Below, we elaborate on how to uniformly sample a neighbor graph $b$ surrounding $a$ in the structural distances, i.e., $(\alpha, \beta) = (1, 0)$.

Let $\mathbb{A} = \{0, 1\}^{n^2}$ be the adjacency matrix space, where $n$ is the number of nodes. $\mathbb{A}$ is a finite-dimensional compact normed vector space, so all the norms are equivalent. Thus, all the induced $\{L^p\}_p$ distances are equivalent. Without loss of generality, we can consider the $L^1$ loss, which exactly corresponds to the *Hamming* distance:

$$d^1([G, X], [\tilde{G}, \tilde{X}]) = \sum_{i \leq j} |\mathbf{A}^{i,j} - \tilde{\mathbf{A}}^{i,j}|,$$

where $\mathbf{A}, \tilde{\mathbf{A}}$ corresponds to the adjacency matrix of the graphs $G, \tilde{G}$ respectively. Since we consider undirected graphs, the Hamming distance takes only discrete values in $\{0, \ldots, \frac{n(n+1)}{2}\}$. In Lemma 4.2, we provide a lower bound for the size of CRF neighbors $\mathcal{N}^{CRF}(a) = \{b \in V^{CRF}|(a, b) \in E^{CRF}\}$ for any GNN input $a \in V^{CRF}$. In Appendix A, we present the proof of Lemma 4.2 and we empirically analyze the asymptotic behavior of this lower bound, motivating thus the need for sampling strategies to reduce the size of the CRF.

**Lemma 4.2.** *For any integer $r$ in $\{0, \ldots, \frac{n(n+1)}{2}\}$, the number of CRF neighboors $|\mathcal{N}^{CRF}(a)|$ for any $a \in V^{CRF}$, i.e., the set of graphs $[\tilde{G}, \tilde{X}]$ with a Hamming distance smaller or equal than $r$, have the following lower bound:*

$$\forall a \in V^{CRF}, \quad \frac{1}{\sqrt{4n(n+1)\epsilon(1-\epsilon)}} \cdot 2^{H(\epsilon)n(n+1)/2} \leq \left|\mathcal{N}^{CRF}(a)\right|, \tag{11}$$

*where $0 \leq \epsilon = \frac{2r}{n(n+1)} \leq 1$ and $H(\cdot)$ is the binary entropy function:*

$$H(\epsilon) = -\epsilon \log_2(\epsilon) - (1-\epsilon)\log_2(1-\epsilon).$$

To sample from $\mathcal{B}([G, X], r)$, we use a *Stratified Sampling* strategy. First, we sample a distance value $d$ from $\{0, 1, \ldots, n^2\}$. To do so, we partition $\mathcal{B}([G, X], r)$ with respect to their distance to the original adjacency matrix $A$:

$$\begin{cases} S_d(A) = \{A \in \mathbb{A}|d^1(A, \tilde{A}) = d\}, \\ \mathcal{B}([G, X], r) = \bigcup_{d \leq r} S_d, \\ \forall d \neq d', S_d \cap S_{d'} = \varnothing. \end{cases}$$

To each distance value $d$, we assign the portion of graphs covered by $S_d(A)$ in $\mathbb{A}$ as:

$$\forall d \in \{0, \ldots, r\}, p(d) = \binom{r}{d}\frac{1}{R} \qquad \text{where} \qquad R = \sum_{d=0}^{r} \binom{r}{d} = 2^r.$$

Second, we uniformly select $d$ positions in the adjacency matrix to be modified and change the element of the $d$ positions by changing the value $A^{i,j}$ to $1 - A^{i,j}$. To uniformly sample neighbor graphs when dealing with features-based distances for graphs, we used the sampling strategy of Abbahaddou et al. (2024).

Our RobustCRF is an attack-independent and model-agnostic robustness approach based on uniform sampling without requiring any training. Therefore, it can be used to enhance GNNs' robustness

Table 1: Attacked classification accuracy ($\pm$ standard deviation) of the GCN, the baselines and the proposed RobustCRF on different benchmark node classification datasets after the features based attack application. ① test accuracy on the original features, ② test accuracy on the perturbed features.

| | Attack | Model | Cora | CiteSeer | PubMed | CS | Texas |
|---|---|---|---|---|---|---|---|
| ① | Clean | GCN | 80.66 (0.41) | 70.37 (0.53) | 78.16 (0.67) | 89.15 (2.06) | 51.35 (18.53) |
| | | RGCN | 77.64 (0.52) | 69.88 (0.47) | 75.58 (0.65) | **92.05 (0.72)** | 51.62 (13.91) |
| | | GCORN | 77.83 (2.33) | **71.68 (1.54)** | 76.03 (1.29) | 88.94 (1.86) | 59.73 (4.90) |
| | | NoisyGCN | **81.04 (0.74)** | 70.36 (0.79) | 78.13 (0.53) | 91.47 (0.92) | 48.91 (20.02) |
| | | RobustCRF | 80.63 (0.38) | 70.30 (0.43) | **78.20 (0.24)** | 88.16 (3.41) | **61.08 (5.01)** |
| ② | Random ($\psi = 0.5$) | GCN | 77.88 (0.90) | 66.65 (1.00) | 73.60 (0.75) | 88.92 (2.04) | 46.49 (15.75) |
| | | RGCN | 67.61 (0.80) | 59.76 (1.01) | 61.93 (1.18) | 90.74 (1.08) | 43.51 (10.22) |
| | | GCORN | 76.28 (1.96) | 67.82 (2.18) | 72.35 (1.43) | 88.31 (2.10) | 60.00 (4.95) |
| | | NoisyGCN | 78.59 (1.09) | 66.83 (0.98) | 73.60 (0.58) | **91.04 (0.85)** | 48.91 (19.29) |
| | | RobustCRF | **78.28 (0.68)** | **68.23 (0.57)** | **74.37 (0.47)** | 88.30 (3.25) | **57.30 (4.32)** |
| | PGD | GCN | 76.38 (0.72) | 67.57 (0.77) | 74.86 (0.65) | 86.90 (1.91) | 52.97 (19.19) |
| | | RGCN | 68.45 (0.97) | 64.63 (0.82) | 73.35 (0.81) | **90.76 (0.68)** | 60.81 (10.27) |
| | | GCORN | 73.32 (2.19) | **69.05 (2.50)** | 74.49 (1.13) | 87.07 (2.96) | 59.73 (4.90) |
| | | NoisyGCN | 76.29 (1.69) | 67.09 (1.50) | 75.04 (0.53) | 88.79 (0.85) | 52.97 (19.11) |
| | | RobustCRF | **76.41 (0.70)** | 67.90 (0.63) | **75.17 (0.91)** | 85.60 (2.75) | **62.16 (4.83)** |

against unknown attack distributions. In Section 5, we will experimentally validate this theoretical insight for the node classification task and demonstrate that RobustCRF has a good trade-off between robustness and clean accuracy, i.e., the model's initial performance on clean un-attacked dataset. Moreover, the approach of our work and these baselines are fundamentally different, since our RobustCRF is post-hoc, we can also use the baselines in combination with our proposed RoustCRF approach to draw even more robust predictions. We report the results of this experiment in Table 2.

*Remark* 4.3. If we set the value of $\sigma$ to 0, the number of iterations to 1, and all the similarity coefficients to 1, this scheme corresponds to the *standard randomized smoothing* with the uniform distribution. In this setting, we do not take into account the original classification task, which causes a huge drop in the clean accuracy. Therefore, RobustCRF is a generalization of the randomized smoothing that gives a better trade-off between accuracy and robustness.

## 5 EXPERIMENTAL SETUP

In this section, we shift from theoretical exploration to practical validation by evaluating the effectiveness of RobustCRF on real-world benchmark datasets. We begin by detailing the experimental setup employed, followed by a thorough presentation and analysis of the results. Our primary experimental objective is to assess how well the proposed method enhances the robustness of a trained GNN.

### 5.1 EXPERIMENTAL SETUP

**Datasets.** For our experiments, we focus on node classification within the general perspective of node representation learning. We use the citation networks Cora, CoraML, CiteSeer, and PubMed (Sen et al., 2008). We additionally consider the co-authorship network CS (Shchur et al., 2018) and the blog and citation graph PolBlogs (Adamic & Glance, 2005), and the non-homophilous dataset Texas (Lim et al., 2021). More details and statistics about the datasets can be found in Table 3. For the CS dataset, we randomly selected 20 nodes from each class to form the training set and 500/1000 nodes for the validation and test sets (Yang et al., 2016). For all the remaining datasets, we adhere to the public train/valid/test splits provided by the datasets.

**Implementation Details.** We used the PyTorch Geometric (PyG) open-source library, licensed under MIT (Fey & Lenssen, 2019). Additionally, for adversarial attacks in this study, we used the DeepRobust package [1]. The experiments were conducted on an RTX A6000 GPU. For the structure-based CRF, we leveraged the sampling strategy detailed in Section 4. The set of hyperparameters for each dataset can be found in Appendix C. We compute the similarity $g_{ab}$ between two inputs $a = [G, X]$ and $b = [\tilde{G}, \tilde{X}]$ using the Cosine Similarity for the features based attacks, namely

---

[1] https://github.com/DSE-MSU/DeepRobust

Table 2: Attacked classification accuracy (± standard deviation) of the baselines when combined with the proposed RobustCRF on different benchmark node classification datasets after the features based attack application.

| Attack | Model | Cora | CiteSeer | PubMed | Texas |
|---|---|---|---|---|---|
| Clean | RGCN | 77.64 (0.52) | **69.88 (0.47)** | **75.58 (0.65)** | 51.62 (13.91) |
| | RGCN w/ RobustCRF | **77.70 (0.46)** | 69.84 (0.39) | 75.50 (0.60) | **52.16 (14.20)** |
| | GCORN | 77.83 (2.33) | 71.68 (1.54) | 76.03 (1.29) | 59.73 (4.90) |
| | GCORN w/ RobustCRF | **78.50 (1.17)** | **71.72 (1.46)** | **76.13 (1.08)** | **59.77 (4.68)** |
| | NoisyGCN | 81.04 (0.74) | **70.36 (0.79)** | 78.13 (0.53) | 48.91 (20.02) |
| | NoisyGCN w/ RobustCRF | **81.07 (0.70)** | 70.20 (0.85) | **78.90 (0.46)** | **49.18 (19.59)** |
| Random ($\psi = 0.5$) | RGCN | 67.61 (0.80) | 59.76 (1.01) | 61.93 (1.18) | 43.51 (10.22) |
| | RGCN w/ RobustCRF | **69.08 (0.73)** | **60.04 (1.01)** | **63.05 (0.88)** | **45.05 (9.99)** |
| | GCORN | 76.28 (1.96) | 67.82 (2.18) | **72.35 (1.43)** | **60.00 (4.95)** |
| | GCORN w/ RobustCRF | **77.21 (1.17)** | **68.94 (3.00)** | 72.29 (1.29) | 59.18 (3.71) |
| | NoisyGCN | 78.59 (1.09) | 66.83 (0.98) | 73.60 (0.58) | **48.91 (19.29)** |
| | NoisyGCN w/ RobustCRF | **81.07 (0.99)** | **67.04 (1.38)** | **74.18 (0.98)** | 45.94 (14.54) |
| PGD | RGCN | 68.45 (0.97) | **64.63 (0.82)** | 73.35 (0.81) | **60.81 (10.27)** |
| | RGCN w/ RobustCRF | **68.46 (0.93)** | 64.59 (0.84) | **73.47 (0.72)** | 58.10 (11.09) |
| | GCORN | 73.32 (2.19) | 69.05 (2.50) | 74.49 (1.13) | 59.73 (4.90) |
| | GCORN w/ RobustCRF | **73.65 (1.55)** | **69.09 (2.57)** | **74.59 (0.90)** | **60.27 (4.68)** |
| | NoisyGCN | 76.29 (1.69) | 67.09 (1.50) | 75.04 (0.53) | 52.97 (19.11) |
| | NoisyGCN w/ RobustCRF | **76.48 (1.65)** | **67.21 (1.35)** | **75.39 (0.45)** | **53.51 (19.07)** |

CosSim$(X, \tilde{X})$, while for the structural attacks, we use the prior distribution $g_{ab} = \binom{r}{d}\frac{1}{2^r}$, where $d$ is the value of the Hamming distance between the original graph $a$ and its sampled neighbor $b$. We note that our code is provided in the supplementary materials and will be made public upon publication.

**Attacks.** We evaluate RobustCRF via the Attack Success Rate (ASR), the percentage of attack attempts that produce successful adversarial examples. For the feature-based attacks, we consider two main types: **(1)** we first consider a random attack which consists of injecting Gaussian noise $\mathcal{N}(0, \mathbf{I})$ to the features with a scaling parameter $\psi = 0.5$; **(2)** we have additionally used the white-box Proximal Gradient Descent (Xu et al., 2019a), which is a gradient-based approach to the adversarial optimization task for which we set the perturbation rate to 15%. For the structural perturbations, we evaluated RobustCRF using the "Dice" (Zügner & Günnemann, 2019) adversarial attack in a black-box setting, where we consider a surrogate model. For this setting, we used an attack budget of 10% (the ratio of perturbed edges).

**Baseline Models.** When dealing with feature-based attacks, we compare RobustCRF with the vanilla GCN (Kipf & Welling, 2017), the feature-based defense method RobustGCN (RGCN) (Zhu et al., 2019), NoisyGNN (Ennadir et al., 2024), and GCORN (Abbahaddou et al., 2024). For the structural attacks, we included RobustGCN (RGCN) and other baselines such as GNN-Jaccard (Wu et al., 2019a), GNN-SVD (Entezari et al., 2020), GNNGuard (Zhang & Zitnik, 2020) and GOOD-AT (Li et al., 2024). For all the models, we used the same number of layers $T = 2$, and with a hidden dimension of 16. The models were trained using the cross-entropy loss function with the Adam optimizer (Kingma & Ba, 2014), the number of epochs $N_{epochs} = 300$, and learning rate 0.01 were kept similar for the different approaches across all experiments. To reduce the impact of random initialization, we repeated each experiment 10 times and used the train/validation/test splits provided with the datasets when evaluating against the feature-based attacks, c.f. Table 1. When evaluating against the structural attacks, c.f. Table 4, we used the split strategy of (Zügner et al., 2018), i.e., we select the largest connected components of the graph and use 10%/10%/80% nodes for training/validation/test.

## 5.2 EXPERIMENTAL RESULTS

**Worst-Case Adversarial Evaluation.** We now analyze the results of RobustCRF for the node classification task. We additionally compared our approach with baselines on the large dataset OGBN-Arxiv. We report all the results for the feature and structure-based adversarial attacks, respectively, in Tables 1, 4, and 8. The results demonstrate that the performance of the GCN is significantly impacted when subject to adversarial attacks of varying strategies. In contrast, the

Table 3: Statistics of the node classification datasets used in our experiments.

| Dataset | #Features | #Nodes | #Edges | #Classes |
|---|---|---|---|---|
| Cora | 1,433 | 2,708 | 5,208 | 7 |
| CoraML | 300 | 2,995 | 8,226 | 7 |
| CiteSeer | 3,703 | 3,327 | 4,552 | 6 |
| PubMed | 500 | 19,717 | 44,338 | 3 |
| CS | 6,805 | 18,333 | 81,894 | 15 |
| PolBlogs | - | 1,490 | 19,025 | 2 |
| Texas | 1,703 | 183 | 309 | 5 |
| Ogbn-arxiv | 128 | 31,971 | 71,669 | 40 |

Table 4: Attacked classification accuracy (± standard deviation) of the GCN, the baselines and the proposed RobustCRF on different benchmark node classification datasets after the structural attack application. ① test accuracy on the original structure, ② test accuracy on the perturbed structure.

| | Attack | Model | Cora | CoraML | CiteSeer | PolBlogs |
|---|---|---|---|---|---|---|
| ① | Clean | GCN | 83.42 (1.00) | 85.60 (0.40) | 70.66 (1.18) | 95.16 (0.64) |
| | | RGCN | 83.46 (0.53) | 85.61 (0.61) | 72.18 (0.97) | 95.32 (0.76) |
| | | GCNGuard | **83.72 (0.67)** | 85.54 (0.42) | 73.18 (2.36) | 95.07 (0.51) |
| | | GCNSVD | 77.96 (0.61) | 81.29 (0.51) | 68.16 (1.15) | 93.80 (0.73) |
| | | GCNJaccard | 82.20 (0.67) | 84.85 (0.39) | **73.57 (1.21)** | 51.81 (1.49) |
| | | GOOD-AT | 83.43 (0.11) | 84.87 (0.15) | 72.80 (0.45) | 94.85 (0.52) |
| | | RobustCRF | 83.52 (0.04) | **85.69 (0.09)** | 72.16 (0.30) | **95.40 (0.24)** |
| ② | Dice | GCN | 81.87 (0.73) | 83.34 (0.60) | 71.76 (1.06) | 87.14 (0.86) |
| | | RGCN | 81.27 (0.71) | 83.89 (0.51) | 69.45 (0.92) | 87.35 (0.76) |
| | | GCNGuard | 81.63 (0.74) | 83.72 (0.49) | 71.87 (1.19) | 86.98 (1.26) |
| | | GCNSVD | 75.62 (0.61) | 79.13 (1.01) | 66.10 (1.29) | 88.50 (0.97) |
| | | GCNJaccard | 80.67 (0.66) | 82.88 (0.58) | **72.32 (1.10)** | 51.81 (1.49) |
| | | GOOD-AT | 82.21 (0.56) | 84.16 (0.36) | 71.43 (0.28) | 90.93(0.38) |
| | | RobustCRF | **82.44 (0.41)** | **84.71 (0.32)** | 71.48 (0.15) | **90.46 (0.25)** |

proposed RobustCRF approach shows a substantial improvement in defense against these attacks compared to other baseline models. Furthermore, in contrast to some other benchmarks, RobustCRF offers an optimal balance between robustness and clean accuracy. Specifically, the proposed approach effectively enhances the robustness against adversarial attacks while maintaining high accuracy on non-attacked, clean datasets. This latter point makes RobustCRF particularly advantageous, as it enhances the model's defenses without compromising its performance on downstream tasks. In Table 2, we report the performance of the baselines when combined with RobustCRF. As noticed, for most of the cases, we further enhace the robustness of the baselines when using RobustCRF.

**Time and Complexity.** We study the effect of the number of iterations $K$ and the number of samples $L$ on the inference time. In Appendix D, we report the average time needed to compute the RobustCRF inference. The results validate the intuitive fact that the inference time grows by increasing $K$ and $L$. We recall that we need to use the model $L^K$ times in the CRF inference.

## 6 CONCLUSION

This work addresses the problem of adversarial defense at the inference stage. We propose a model-agnostic, post-hoc approach using Conditional Random Fields (CRFs) to enhance the adversarial robustness of pre-trained models. Our method, RobustCRF, operates without requiring knowledge of the underlying model and necessitates no post-training or architectural modifications. Extensive experiments on multiple datasets demonstrate RobustCRF's effectiveness in improving the robustness of Graph Neural Networks (GNNs) against both structural and node-feature-based adversarial attacks, while maintaining a balance between attacked and clean accuracy, typically preserving their performance on clean, un-attacked datasets, which makes RobustCRF the best trade-off between robustness and clean accuracy.

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

# A   PROOF OF LEMMA 4.1

By solving the system of equations in equation 4, we can get the optimal distribution $Q^*$ as follows:

$$\forall a \in V^{CRF}, \ \ Q(\tilde{Y}_a) \propto \exp\left\{ \mathbb{E}_{-a}\left[ log\ P\left( \tilde{Y}|Y, V^{CRF}, E^{CRF}\right)\right]\right\}.$$

*Proof.* The $\mathcal{KL}$-divergence includes the true posterior $P(\tilde{Y}|Y, V^{CRF}, E^{CRF})$ which is exactly the unknown value. We can rewrite the $\mathcal{KL}$-divergence as:

$$\mathcal{KL}\left(Q|P\right) = \int Q(\tilde{Y}) \log \frac{Q(\tilde{Y})}{P(\tilde{Y}|Y, V^{CRF}, E^{CRF})} d\tilde{Y}$$

$$= \int Q(\tilde{Y}) \log \frac{Q(\tilde{Y})P(Y, V^{CRF}, E^{CRF})}{P(\tilde{Y}, Y, V^{CRF}, E^{CRF})} d\tilde{Y}$$

$$= \int Q(\tilde{Y}) \left( \log\ P(Y, V^{CRF}, E^{CRF}) + \log \frac{Q(\tilde{Y})}{P(\tilde{Y}, Y, V^{CRF}, E^{CRF})} \right) d\tilde{Y}$$

$$= \log\ P(Y, V^{CRF}, E^{CRF}) \int Q(\tilde{Y})d\tilde{Y} - \int Q(\tilde{Y}) \log \frac{P(\tilde{Y}, Y, V^{CRF}, E^{CRF})}{Q(\tilde{Y})} d\tilde{Y}.$$

Since $\int Q(\tilde{Y})d\tilde{Y} = 1$, we conclude that:

$$\mathcal{KL}\left(Q|P\right) = \log\ P(Y, V^{CRF}, E^{CRF}) - \int Q(\tilde{Y}) \log \frac{P(\tilde{Y}, Y, V^{CRF}, E^{CRF})}{Q(\tilde{Y})} d\tilde{Y}.$$

We are minimizing the $\mathcal{KL}-$divergence over $Q$, therefore the term $log\ P(Y, V^{CRF}, E^{CRF})$ can be ignored. The second term is the *This is the negative ELBO*. We know that the $\mathcal{KL}-$divergence is not negative. Thus, $\log\ P(Y, V^{CRF}, E^{CRF}) \geq \text{ELBO}(Q)$ justifying the name *Evidence lower bound (ELBO)*.

Therefore, the main objective is to optimize the ELBO in the mean field variational inference, i.e., choose the variational factors that maximize ELBO:

$$\text{ELBO}(Q) =$$

$$\int Q(\tilde{Y}) \log \frac{P(\tilde{Y}, Y, V^{CRF}, E^{CRF})}{Q(\tilde{Y})} d\tilde{Y} = \mathbb{E}_Q\left[ \log\ P(\tilde{Y}, Y, V^{CRF}, E^{CRF})\right] - \mathbb{E}_Q\left[ \log\ Q(\tilde{Y})\right].$$

$$(12)$$

We will employ *coordinate ascent inference*, where we iteratively optimize each variational distribution while keeping the others constant.

We assume that the set of CRF nodes is finite, i.e., $\left| V^{CRF} \right| < \infty$, which is a realistic assumption if we consider the set of all GNN inputs used during inference. If $\left| V^{CRF} \right| = m$, we can order the elements $V^{CRF}$ in a a specific order $i = 1, \ldots, m$. Thus, using the chain rule, we decompose the probability $P(\tilde{Y}, Y, V^{CRF}, E^{CRF})$ as follows:

$$P(\tilde{Y}, Y, V^{CRF}, E^{CRF}) = P(\tilde{Y}_{1:m}, Y_{1:m}, V^{CRF}, E^{CRF})$$

$$= P(Y_{1:m}, V^{CRF}, E^{CRF}) \prod_{i=1}^{m} P(\tilde{Y}_i|Y_{1:(i-1)}, V^{CRF}, E^{CRF}).$$

Using the independence of the mean field approximation, we also have:

$$\mathbb{E}_Q\left[ \log\ Q(\tilde{Y})\right] = \sum_{i=1}^{m} \mathbb{E}_{Q_j}\left[ \log\ Q(\tilde{Y}_j)\right]$$

Now, we have the expression of the two terms appearing in ELBO in equation 12:

$$\text{ELBO}(Q) = \log\ P(Y_{1:m}, V^{CRF}, E^{CRF}) + \sum_{i=1}^{m} \mathbb{E}_Q\left[ \log\ P(\tilde{Y}_i|Y_{1:(i-1)}, V^{CRF}, E^{CRF})\right] - \mathbb{E}_{Q_j}\left[ \log\ Q(\tilde{Y}_i)\right]$$

The above decomposition is valid for any ordering of the GNN inputs. Thus, for a fixed GNN input $a \in V^{CRF}$, if we consider $a$ as the last variable $m$ of the list, we can consider the ELBO as a function of $Q(\tilde{Y}_a) = Q(\tilde{Y}_m)$:

$$\text{ELBO}(Q(\tilde{Y}_a)) = \text{ELBO}(Q(\tilde{Y}_k))$$

$$= \mathbb{E}_Q \left[ \log\ P(\tilde{Y}_m | Y_{1:(m-1)}, V^{CRF}, E^{CRF}) \right] - \mathbb{E}_{Q_m} \left[ \log\ Q(\tilde{Y}_m) \right]\ +\ const$$

$$= \int Q(\tilde{Y}_m) \mathbb{E}_{Q_{\neq m}} \left[ \log\ P(\tilde{Y}_m | Y_{\neq m}, V^{CRF}, E^{CRF}) \right] d\tilde{Y}_m - \int Q(\tilde{Y}_m) log\ Q(\tilde{Y}_m) d\tilde{Y}_m$$

$$= \int Q(\tilde{Y}_a) \mathbb{E}_{Q_{\neq a}} \left[ \log\ P(\tilde{Y}_a | Y_{\neq a}, V^{CRF}, E^{CRF}) \right] d\tilde{Y}_a - \int Q(\tilde{Y}_a) \log\ Q(\tilde{Y}_a) d\tilde{Y}_a,$$

where $\neq m$ means all indices except the $m^{th}$. Now, we take the derivative of the ELBO with respect to $Q(\tilde{Y}_a)$:

$$\frac{d\ \text{ELBO}}{dQ(\tilde{Y}_a)} = \mathbb{E}_{Q_{\neq a}} \left[ \log\ P(\tilde{Y}_a | Y_{\neq a}, V^{CRF}, E^{CRF}) \right] - \log\ Q(\tilde{Y}_a) - 1 = 0.$$

Therefore,

$$\forall a \in V^{CRF},\ \ Q(\tilde{Y}_a) \propto \exp \left\{ \mathbb{E}_{-a} \left[ \log\ P \left( \tilde{Y} | Y, V^{CRF}, E^{CRF} \right) \right] \right\}.$$

$\square$

# B   THE NUMBER OF CRF NEIGHBORS

## B.1   PROOF OF LEMMA 4.2

For any integer $r$ in $\{0, \ldots, \frac{n(n+1)}{2}\}$, the number of CRF neighboors $\left|\mathcal{N}^{CRF}(a)\right|$ for any $a \in V^{CRF}$, i.e., the set of graphs $[\tilde{G}, \tilde{X}]$ with a Hamming Distance smaller or equal than $r$, have the following lower bound:

$$\forall a \in V^{CRF}, \quad \frac{1}{\sqrt{4N(N+1)\epsilon(1-\epsilon)}} \cdot 2^{H(\epsilon)N(N+1)/2} \leq \left|\mathcal{N}^{CRF}(a)\right|, \tag{13}$$

where $0 \leq \epsilon = \frac{2r}{n(n+1)} \leq 1$ and $H(\cdot)$ is the he binary entropy function:

$$H(\epsilon) = -\epsilon \log_2(\epsilon) - (1-\epsilon) \log_2(1-\epsilon).$$

*Proof.* We use Stirling's formula;

$$\forall s, \quad s! = \sqrt{2\pi s} s^s e^{-s} \exp\left(\frac{1}{12s} - \frac{1}{360s^3} + \ldots\right). \tag{14}$$

Thus, for $r$ in $\{0, \ldots, \frac{n(n+1)}{2}\}$ and $L = \frac{n(n+1)}{2}$, we write

$$\binom{\frac{n(n+1)}{2}}{r} = \binom{L}{r} \tag{15}$$

$$= \frac{L}{r!(L-r)!} \tag{16}$$

$$\geq \frac{\sqrt{2\pi L} L^L e^{-L} \exp\left[-1/12r - 1/12(L-r)\right]}{\sqrt{2\pi r} r^r e^{-r} \sqrt{2\pi(L-r)}(L-r)^{(L-r)} e^{-(L-r)}}. \tag{17}$$

For $L \geq 4$, for any $r \in \{1, \ldots, L\}$, we always have $L - r \geq 3$ or $r \geq$, thus,

$$\frac{1}{12r} + \frac{1}{12(L-r)} \leq \frac{1}{12} + \frac{1}{36} = \frac{1}{9}. \tag{18}$$

Therefore,

$$\exp\left(-\frac{1}{12r} - \frac{1}{12(L-r)}\right) \geq \frac{1}{12} + \frac{1}{36} = e^{-1/9} \geq \frac{1}{2}\sqrt{\pi}. \tag{19}$$

We can then therefore derive a lower bound for $\binom{\frac{n(n+1)}{2}}{r}$ :

$$\binom{\frac{n(n+1)}{2}}{r} = \binom{L}{r} \tag{20}$$

$$\geq \frac{\sqrt{2\pi L} L^L e^{-L} \frac{1}{2}\sqrt{\pi}}{\sqrt{2\pi r} r^r e^{-r} \sqrt{2\pi(L-r)}(L-r)(L-r)e^{-(L-r)}} \tag{21}$$

$$= \frac{\sqrt{L} L^L}{\sqrt{8r} r^r \sqrt{(L-r)}(L-r)^{(L-r)}} \tag{22}$$

$$= \sqrt{\frac{L}{8r(L-r)}} \frac{L^L}{r^r (L-r)^{(L-r)}} \tag{23}$$

$$= \frac{1}{\sqrt{8L\epsilon(1-\epsilon)}} \frac{1}{\epsilon^r (1-\epsilon)^{L-r}} \tag{24}$$

$$= \frac{1}{\sqrt{8L\epsilon(1-\epsilon)}} \epsilon^{-L\epsilon} (1-\epsilon)^{L(1-\epsilon)} \tag{25}$$

$$= \frac{1}{\sqrt{8L\epsilon(1-\epsilon)}} 2^{LH(\epsilon)}. \tag{26}$$

For $d$ in $\{0, \ldots, \frac{n(n+1)}{2}\}$, the number of possible graphs with a Hamming distance equal to $d$ from a graph $a = [G, x]$ is $\binom{L}{d}$. Thus,

$$\left|\mathcal{N}^{CRF}(a)\right| = \sum_{d=0}^{r} \binom{L}{d} \tag{27}$$

$$\geq \binom{L}{r} \tag{28}$$

$$\geq \frac{1}{\sqrt{8L\epsilon(1-\epsilon)}} 2^{LH(\epsilon)} \tag{29}$$

$$= \frac{1}{\sqrt{4N(N+1)\epsilon(1-\epsilon)}} \cdot 2^{H(\epsilon)N(N+1)/2}. \tag{30}$$

$\square$

### B.2 EMPIRICAL INVESTIGATION OF THE LOWER BOUND

We empirically investigate the evolution of the lower-bound stated in Lemma 4.2 as a function of ratio $\epsilon = \epsilon(r)$. As noticed, the number of neighbors increases exponentially as the radius increases. This motivates the need for sampling strategies to reduce the size of the CRF.

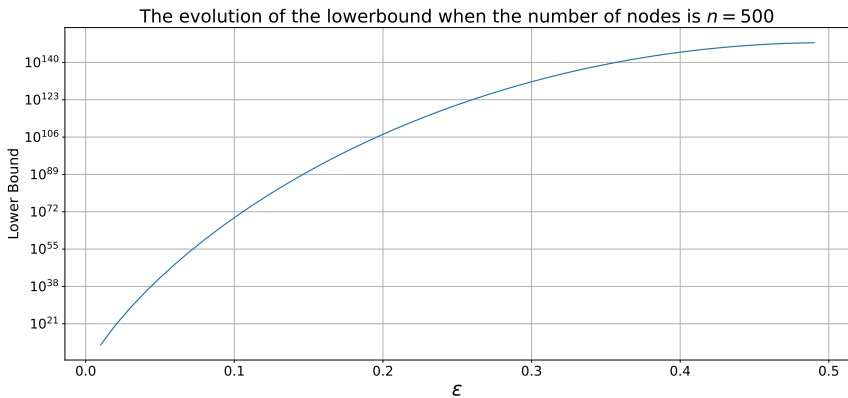

Figure 2: The effect of the radius on the lower bound stated in Lemma 4.2

## C    THE SET OF HYPERPARAMETERS USED TO CONSTRUCT THE CRF

In Tables 5 and 6, we present the hyperparameters used to run the inference of RobustCRF for each dataset. The number of iterations was fixed to 2. $p_r$ and we compute the radius as a floor function, i.e.,$r = \lfloor p_r \times m \rfloor$, where $m = |E|$ is the number of existing edges in the original graph.

Table 5: The optimal RobustCRF's hyperparameters for each dataset when dealing with the feature-based attacks.

| Hyperparameter | ora | CiteSeer | PubMed | CS |
|---|---|---|---|---|
| $r$ | 0.1 | 0.9 | 0.3 | 0.3 |
| $\sigma$ | 0.9 | 0.8 | 0.9 | 0.5 |

Table 6: The optimal RobustCRF's hyperparameters for each dataset when dealing with the structure-based attacks.

| Hyperparameters | Cora | CoraML | CiteSeer | PolBlogs |
|---|---|---|---|---|
| $p_r$ | 0.02 | 0.02 | 0.04 | 0.005 |
| $\sigma$ | 0.05 | 0.2 | 0.05 | 0.05 |

# D   TIME AND COMPLEXITY OF ROBUSTCRF

In Table 7, we present the average time (in seconds) required for RobustCRF inference. As observed, the inference time increases exponentially with larger values of $K$ and $L$. Nevertheless, empirical evidence suggests that a small number of samples is sufficient to improve GNN robustness. In our experiments, we specifically used $L = 5$ samples and set the number of iterations to 2.

Table 7: Inference time for different values of the number of iterations/samples.

| Num Samples $L$ | 0 Iter | 1 Iter | 2 Iter |
|---|---|---|---|
| 5 | $0.26 \pm 0.52$ | $1.99 \pm 0.54$ | $16.40 \pm 2.04$ |
| 10 | $0.20 \pm 0.41$ | $3.24 \pm 0.47$ | $58.28 \pm 1.15$ |
| 20 | $0.22 \pm 0.44$ | $5.86 \pm 0.53$ | $224.86 \pm 0.71$ |

# E   RESULS ON OGBN-ARXIV

Table 8: Attacked classification accuracy ($\pm$ standard deviation) of the GCN, a baseline and the proposed RobustCRF on the OGBN-Arxiv dataset.

| Dataset | GCN | NoisyGCN | RobustCRF |
|---|---|---|---|
| Clean | **60.41 (0.15)** | 59.97 (0.11) | 60.28 (0.15) |
| Random | 58.97 (0.24) | 58.71 (0.10) | **59.03 (0.26)** |
| PGD | 50.24 (0.42) | **50.26 (0.37)** | 50.10 (0.56) |

