# OpenReview forum: "Post-Hoc Robustness Enhancement in Graph Neural Networks with Conditional Random Fields"
_ICLR.cc/2025/Conference — ICLR 2025 Conference Withdrawn Submission_

### Official Review · Reviewer_rT7V · 2024-10-26

**Soundness:** 1
**Presentation:** 2
**Contribution:** 2
**Rating:** 3
**Confidence:** 4

**Summary:**

This study introduces RobustCRF, a post-hoc method designed to enhance the robustness of Graph Neural Networks (GNNs) specifically at the inference stage. Unlike most existing defense techniques that focus on training adjustments, RobustCRF leverages Conditional Random Fields in a model-agnostic manner, requiring no prior knowledge of the GNN architecture. The approach is validated on benchmark node classification datasets, demonstrating its effectiveness across various models in improving resilience against adversarial attacks.

**Strengths:**

1. Model-Agnostic Robustness: The proposed method, RobustCRF, enhances the robustness of Graph Neural Networks (GNNs) without requiring modifications to the model's underlying architecture, making it versatile and adaptable to various GNN models.

2. Theoretical Foundation and Complexity Reduction: The approach is grounded in a comprehensive theoretical analysis, which strengthens the understanding and reliability of the proposed method. Additionally, it includes sampling techniques that reduce the computational complexity, making it more efficient.

3. Post-Hoc Enhancement: As a post-hoc technique, RobustCRF can be applied after model training, allowing for flexibility and ease of implementation across different GNNs, as it doesn't require prior modifications to the model.

**Weaknesses:**

1. The contribution of this work is quite limited. Although it provides a theoretical framework of CRF for designing GNN architecture, the authors heuristically define the potential functions $\phi_a$ and $\phi_{ab}$ without any theoretical guarantee or intuition from the CRF perspective. These two terms actually follow the robustness criteria of most existing works.

2. The experimental evaluation is not comprehensive:
   * **Baselines**: The selected baselines are insufficient. How does RobustCRF compare to RUNG [1], ElasticGNN [2], and SoftMedian [3]?
   * **Attacks**: The main evaluation focuses on feature-based attacks, which are not effective for graph-based models. For structure-based models, the authors only compare the black-box Dice attack, which is not a commonly used attack in this field. The authors should evaluate the model under topology PGD attack, metattack, nettack, or node injection attack.
   * **Performance**: The models show very limited improvement over other baselines, or even underperform them.

[1] Robust Graph Neural Networks via Unbiased Aggregation

[2] Elastic Graph Neural Networks

[3] Robustness of Graph Neural Networks at Scale

[4] Graph Neural Networks Inspired by Classical Iterative Algorithms

3. The presentation needs further improvement:
   * **Notation**: The notation is poor and very confusing. The input notation should be consistent. In the paper, $[A,X]$, $[G,X]$, and $\mathcal{G} \times \mathcal{X}$ appear to represent the same thing.
   * **Clarity**: What do $\tilde{G}$, $\tilde{X}$, and $\tilde{Y}$ mean? Sometimes they denote the perturbed input, other times updated predictions, and occasionally they are used to differentiate two inputs. This makes it difficult for readers to follow the paper.
   * **Redundant Information**: Sections 4.1 and 4.2 seem redundant. This information is commonly known in the robustness of GNNs but occupies two subsections.
   * **Table Placement**: Tables are not positioned properly. For example, Tables 1 and 4 should be near the main experiment, and Table 2 (ablation study) should not be placed between the main result tables (Tables 1 and 4).

4. The paper lacks ablation studies to validate the effectiveness of the proposed RobustCRF. For example, the authors should include an ablation study to assess the impact of $\alpha$ in the potential functions, the number of iterations, etc.

**Questions:**

Refer to weaknesses

---

### Official Review · Reviewer_p8uV · 2024-10-29

**Soundness:** 3
**Presentation:** 2
**Contribution:** 3
**Rating:** 5
**Confidence:** 3

**Summary:**

This paper put forward a method that can enhance the robustness of graph neural networks during the inference stage. This method is based on a Conditional Random Field. Extensive experiments show the effectiveness of the proposed method.

**Strengths:**

The paper has the following strengths:
- This paper is well-written and the method is easy to follow.
- The proposed method is model-agnostic and can be applied to many graph neural networks.
- Detailed analysis shows the effectiveness of the proposed method from a theoretical perspective.

**Weaknesses:**

I think the weakness lies in insufficient experiments. Some experiments need to be refined or added.

For example, for the structural perturbations. The chosen attack method is old. Could authors consider using some new SOTA attack method?

Besides, it would be better to have all the experiments in Table 8 look like in Table 1. Besides, for the results in Table 4, why are there no results on the methods with RobustCRF? (E.g., GCNSVD w/RobustCRF, and GCNJaccard w/RobustCRF). Is RobustCRF also compatible with them?

**Questions:**

I have some questions.

The first question is about the evaluation metric. As written in line 457 "We evaluate RobustCRF via the eAttack Success Rate(ASR)," However, in the table caption, it says 'Attacked classification accuracy'. So I am a little confused. Does the &#9312;
 in your table 1 represents clean accuracy and &#9313; represents ASR?

The second question is. I think there are there already some Post-Hoc Robustness Enhancement methods for graph neural networks. Can authors add some experiments on how the defense methods (such as GCNSVD, RGCN, and GCORN) perform when combined with those methods?

---

### Official Review · Reviewer_9EUe · 2024-10-31

**Soundness:** 3
**Presentation:** 3
**Contribution:** 2
**Rating:** 5
**Confidence:** 3

**Summary:**

This paper presents RobustCRF, a post-hoc, model-agnostic method to enhance the robustness of Graph Neural Networks (GNNs) against adversarial attacks during inference. By leveraging a Conditional Random Field (CRF), RobustCRF enforces that similar nodes produce similar outputs, improving robustness without prior knowledge of the GNN architecture. The method includes a sampling strategy to reduce computational complexity and is validated through theoretical and empirical analysis on benchmark datasets, demonstrating its efficiency and applicability across various GNN models.

**Strengths:**

1. The paper is well-written and easy to follow.
2. Proposed framework is theoretical supported.
3. Introducing CRF to GNN robustness is interesting.

**Weaknesses:**

1. It is not clear why, in the clean dataset setting, RobustCRF can achieve comparable or even better performance than GCN. Intuitively, RobustCRF should perform worse than the backbone on a clean dataset. Could the authors provide an explanation for this phenomenon?
2. Since RobustCRF introduces new hyperparameters, could the authors provide a hyperparameter analysis?
3. The performance improvement appears marginal, especially under the PGD attack in Table 1. Could the authors provide more explanation on this?

**Questions:**

please refer to weakness.

---

### Official Review · Reviewer_J5wN · 2024-11-02

**Soundness:** 2
**Presentation:** 2
**Contribution:** 1
**Rating:** 3
**Confidence:** 5

**Summary:**

This work studies the adversarial robustness of Graph Neural Networks during inference (evasion attack) and propose a new defense method RobustCRF basde on Conditional Random Field. The basic idea of this work is based on an assumption that similar inputs, including features and structures, should have similar output, and RobustCRF tries to achieve this by adapting the output.

**Strengths:**

1. This works is easy to understand.

2. The motivation of this work is reasonable.

**Weaknesses:**

The biggest issue with this paper is that its literature review is outdated. It does not cite any papers (except GCORN) in the field of GNN adversarial robustness published after 2023, resulting in the omission of many recent advancements in evasion attack and defense. For example:

>*Missing discussion with adaptive attack.*

Adversarial robustness studies the generalization in the worst-case scenario. [1] provides a detailed explanation of how existing GNN defenses are highly vulnerable to adaptive evasion attacks and propose a set of perturbed graphs as a bare minimum robustness unit test.

Nicholas Carlini has elaborated extensively on this topic. When designing a defense method, ensuring attack agnosticism and assuming the adversary is aware of the defense are better standards for evaluating the robustness of a method. If a method can be bypassed by a very simple trick, how can we argue that it is adversarially robust?

Authors also mentioned the hope that the method would be robust against unknown future attacks. Therefore, assuming robustness in any situation is necessary, which is why considering adaptive attacks is essential when developing defenses.

>*Missing comparison with the current state-of-the-art defenses against evasion attacks, adversarial training [2, 3].*

The experimental setup of adversarial training is same as that of this paper and shares a similar motivation—encouraging the model to be locally invariant so that it maintains the same output within the range of semantically preserving perturbations. Based on the experimental results presented, the robustness of CRFRobust does not show a significant advantage compared to the baselines used in the paper. Therefore, it is highly likely that adversarial training would perform much better than CRFRobust.

>*The attacks used in this work are TOO weak.*

The structural attack baseline used in this work, DICE, is a heuristic attack based on homophily assumption. However, it serves as a very weak attack in this filed. The authors should consider stronger attack settings, similar to those described in [2] and [3].

### References:

[1] Are Defenses for Graph Neural Networks Robust? NeurIPS 2022.

[2] Adversarial Training for Graph Neural Networks, NeurIPS 2023.

[3] Boosting the Adversarial Robustness of Graph Neural Networks: An OOD Perspective, ICLR 2024.

**Questions:**

**1. What is the difference between defense against evasion attack and post-hoc defense.**

Both of them assume that attack happens during the inference phase, so the training procedure is not affected. If we formalize this problem mathematically, there are no differece. Therefore, all the reference work [1, 2, 3] that I mentioned in the previous review comprise the defense or evaluation on evasion attack. It is very necessary to consider them, see reasons below. [1] provide a systematical way to evaluate adaptive robustness of evasion attack; [2] is the first work that make adversarial training work on GNNs, and adversarial training is considered the only way to defense adversarial attack in the filed of vision community.

---

### Note · Authors · 2024-12-04

**Comment:**

We would very much like to thank the reviewers for their insightful reviews and the AC for taking the time to handle our paper. Unfortunately, we were unable to produce a comprehensive, convincing rebuttal during the discussion period and would therefore like to withdraw the paper to not take anymore of your time. We will certainly take the valuable feedback provided into account in future versions of this work. Thank you very much again.

**Withdrawal Confirmation:**

I have read and agree with the venue's withdrawal policy on behalf of myself and my co-authors.